# A Diagnostic Algorithm Based on a Simple Clinical Prediction Rule for the Diagnosis of Cranial Giant Cell Arteritis

**DOI:** 10.3390/jcm10061163

**Published:** 2021-03-10

**Authors:** Michael Czihal, Christian Lottspeich, Christoph Bernau, Teresa Henke, Ilaria Prearo, Marc Mackert, Siegfried Priglinger, Claudia Dechant, Hendrik Schulze-Koops, Ulrich Hoffmann

**Affiliations:** 1Division of Vascular Medicine, Medical Clinic and Policlinic IV, Hospital of the Ludwig-Maximilians-University, 80336 Munich, Germany; christian.lottspeich@med.uni-muenchen.de (C.L.); christoph.bernau@gmx.de (C.B.); teresa.henke@med.uni-muenchen.de (T.H.); ilaria.prearo@med.uni-muenchen.de (I.P.); ulrich.hoffmann@med.uni-muenchen.de (U.H.); 2Interdisciplinary Sonography Center, Medical Clinic and Policlinic IV, Hospital of the Ludwig-Maximilians-University, 80336 Munich, Germany; 3Department of Ophthalmology, Hospital of the Ludwig-Maximilians-University, 80336 Munich, Germany; marc.mackert@med.uni-muenchen.de (M.M.); siegfried.priglinger@med.uni-muenchen.de (S.P.); 4Division of Rheumatology and Clinical Immunology, Medical Clinical and Policlinic IV, Hospital of the Ludwig-Maximilians-University, 80336 Munich, Germany; claudia.dechant@med.uni-muenchen.de (C.D.); hendrik.schulze-koops@med.uni-muenchen.de (H.S.-K.)

**Keywords:** giant cell arteritis, anterior ischemic optic neuropathy, clinical prediction rule, diagnostic algorithm, C-reactive protein, temporal compression sonography, ultrasound

## Abstract

Background: Risk stratification based on pre-test probability may improve the diagnostic accuracy of temporal artery high-resolution compression sonography (hrTCS) in the diagnostic workup of cranial giant cell arteritis (cGCA). Methods: A logistic regression model with candidate items was derived from a cohort of patients with suspected cGCA (*n* = 87). The diagnostic accuracy of the model was tested in the derivation cohort and in an independent validation cohort (*n* = 114) by receiver operator characteristics (ROC) analysis. The clinical items were composed of a clinical prediction rule, integrated into a stepwise diagnostic algorithm together with C-reactive protein (CRP) values and hrTCS values. Results: The model consisted of four clinical variables (age > 70, headache, jaw claudication, and anterior ischemic optic neuropathy). The diagnostic accuracy of the model for discrimination of patients with and without a final clinical diagnosis of cGCA was excellent in both cohorts (area under the curve (AUC) 0.96 and AUC 0.92, respectively). The diagnostic algorithm improved the positive predictive value of hrCTS substantially. Within the algorithm, 32.8% of patients (derivation cohort) and 49.1% (validation cohort) would not have been tested by hrTCS. None of these patients had a final diagnosis of cGCA. Conclusion: A diagnostic algorithm based on a clinical prediction rule improves the diagnostic accuracy of hrTCS.

## 1. Introduction

With an estimated lifetime risk of at 1% for women and 0.5% for men for developing the disease, giant cell arteritis (GCA) is the most common systemic vasculitis [1]. Diagnostic imaging nowadays plays an important role in the diagnosis of GCA. For the diagnostic workup of suspected cranial GCA (cGCA), color duplex sonography (CDS) of the temporal arteries is recommended as the first line imaging test [2,3,4,5]. High-resolution sonography of the cranial arteries has been shown to accurately discriminate patients with and without cGCA [6,7,8]. However, recent evidence suggests that arteriosclerotic wall thickening may impair specificity of this diagnostic test [9,10]. In the absence of a disease-specific biomarker, the C-reactive protein (CRP), offering high diagnostic sensitivity but low specificity, is recommended by current guidelines for the evaluation of suspected GCA [3]. Beyond test accuracy, the predictive value of a diagnostic method is substantially influenced by pre-test probability, which is determined by the prevalence of the disease in the studied population [11]. Therefore, clinical assessment of pre-test probability is of outstanding importance in order to optimize test performance. Current recommendations suggest imaging test interpretation based on clinical pre-test probability [2,3,4]. In clinical practice, assessment of pre-test probability for GCA is usually conducted by implicit (unstructured) clinical judgement, but several clinical prediction rules have been developed and in part validated in order to perform explicit (structured) pre-test probability assessment [12,13,14,15,16,17,18]. To date, none of these clinical prediction rules have gained broad acceptance for clinical use. While the 2018 European League against Rheumatism (EULAR) recommendations for the management of the large vessel vasculitides do not support the use of any prediction rule [3], the 2020 British Society for Rheumatology guideline on diagnosis and treatment of GCA for the first time mentions clinical prediction rules as being potentially useful for assisting clinicians in the estimation of GCA probability [4].

In order to optimize test performance of high-resolution compression sonography (hrTCS) of the temporal arteries in the diagnostic workup of suspected cGCA, the present study aimed at establishing and validating a clinical prediction rule and integrating this rule together with the CRP values in a clinically useful diagnostic algorithm. The diagnostic algorithm was not intended to cover the diagnostic workup of extracranial GCA.

## 2. Materials and Methods

### 2.1. Cohort Characteristics

The study, approved by the local ethics committee (project number: 18-502), was based on two independent cohorts of consecutive patients with suspected cGCA evaluated at a single interdisciplinary vasculitis center [8,10]. All patients in both cohorts underwent a sonographic study of the temporal and axillary arteries, including hrTCS of the temporal arteries. The optimal cut-off of temporal artery wall thickness (sum of the near and far arterial wall) had previously been determined at ≥0.7 mm (sensitivity and specificity of 85% and 95%, respectively) [8]. In the first cohort, published in 2017, diagnosis of cGCA was based on fulfilment of >3 American College of Rheumatology classification criteria and/or a positive temporal artery biopsy (TAB) [8,19]. In the second cohort, published in 2020, a diagnosis of cGCA was established when at least three of the five following criteria were fulfilled: (1) age > 50 years; (2) typical cranial symptoms (new onset, persisting headache, jaw claudication, temporal artery tenderness); (3) unequivocal symptoms of polymyalgia rheumatica; (4) erythrocyte sedimentation rate > 30 mm per 1 h (reference range ≤ 20 mm per one hour) or CRP ≥ 1 mg/dL; (normal range < 0.5 mg/dL); (5) typical hypoechogenic wall thickening (Halo) of the superficial temporal arteries or positive TAB [6,10]. In both cohorts, extracranial GCA was diagnosed based on typical imaging findings in CDS of the axillary arteries (hypoechogenic circumferential wall thickening of ≥1.2 mm) and non-invasive cross-sectional imaging (magnetic resonance imaging or positron emission tomography/computed tomography) [8,10].

### 2.2. Derivation of the Clinical Model

We performed a literature review to identify clinical symptoms and signs predictive for a diagnosis of cGCA [12,13,14,15,16,17,20,21,22,23,24,25,26,27,28,29,30]. We analyzed the prevalence of potential candidate items in the first cohort, comprising 92 patients referred to our institution with suspected GCA (cranial and/or extracranial) between October 2014 and October 2015 [8]. CRP values were available for 87 of 92 patients who were included in the present analysis (26 patients with a final clinical diagnosis of cGCA). Univariate comparisons between patients with and without a final diagnosis of cGCA were made using χ2 test (categorical variables) and Mann–Whitney *U* test (continuous variables). Based on the literature review, results of the univariate analysis, and subject matter expertise, various logistic regression models consisting only of categorical variables were set up. Clinical symptoms and objective ophthalmological findings were considered for modelling, whereas laboratory values and vascular imaging findings were not. The strength of association between selected categorical variables and the diagnosis of cGCA is expressed as logarithmic odds ratio (logOR), with the respective 95% confidence intervals (95% CI). In order to keep the clinical model as simple as possible, continuous variables were not included. For further analysis, the model with the lowest Aikake’s information criterion (AIC) was used. The diagnostic accuracy of this model for discrimination of patients with and without a final diagnosis of cGCA in the derivation cohort was assessed using receiver operator characteristics (ROC) analysis.

### 2.3. Validation of the Clinical Model in an Independent Cohort

Based on the results found in the derivation cohort, we validated the logistic model in the second cohort. This validation cohort consisted of 114 patients referred for sonography of the temporal and axillary arteries, including hrTCS of the temporal arteries as part of the diagnostic workup of acute arterial perfusion disorders of the eye (30 patients with a final diagnosis of cGCA) [10].

### 2.4. Stepwise Diagnostic Algorithm Integrating the Clinical Model, C-Reactive Protein and hrTCS

We integrated the items from the final clinical model rule together with the CRP values and hrTCS (temporal artery wall thickness cut-off of ≥0.7 mm) into a stepwise diagnostic algorithm with the aim of stratification of pre-test probability of cGCA in a low risk vs. a non-low risk category. Based on current evidence, the cut-off of the CRP was set at ≥2.5 mg/dL [12,31,32]. The diagnostic accuracy of the diagnostic algorithm for the diagnosis/exclusion of cGCA in both cohorts was assessed using 2 × 2 contingency tables.

### 2.5. Statistical Analysis

All steps of statistical analysis, as stated above, were performed with the R software for statistical computing (R Development Core Team, Vienna, Austria). Two-sided *p*-values < 0.05 were considered significant. Results for categorical variables are presented as absolute numbers with percentages, and continuous variables are displayed as mean ± standard deviation (SD).

## 3. Results

### 3.1. Derivation of the Clinical Model

The clinical characteristics of patients with (*n* = 26) and without (*n* = 61) a final clinical diagnosis of cGCA in the derivation cohort are compared in Table 1. The specific diagnoses in patients not classified as suffering from cGCA are listed in Appendix A. Logistic regression analysis substantiated the results of the literature review [12,13,14,15,16,17,20,21,22,23,24,25,26,27,28,29,30], showing that a model including jaw claudication (logOR 4.1, 95% CI 1.7–6.7), new onset permanent headache (logOR 3.5, 95% CI 0.6–4.8), age > 70 (logOR 2.2, 95% CI 0.03–4.4), and an ophthalmological diagnosis of anterior ischemic optic neuropathy (AION; unilateral: logOR 2.7, 95% CI 0.6–4.7; bilateral: logOR 3.5, 95% CI 0.07–6.8) discriminated patients with and without cGCA with the highest diagnostic accuracy (AIC 48.6; area under the curve (AUC) 0.96).

### 3.2. Validation of the Logistic Model in an Independent Cohort

The clinical characteristics of patients with (*n* = 30) and without (*n* = 84) a final clinical diagnosis of cGCA in the validation cohort are listed in Table 1. The specific diagnoses in patients not classified as suffering from cGCA are listed in the Appendix A. According to ROC analysis, the logistic model exhibited an area under the curve of 0.92 for correct classification of the final clinical diagnosis (cGCA vs. alternative diagnosis) in the validation cohort. 

### 3.3. Stepwise Diagnostic Algorithm Integrating the Clinical Model, C-Reactive Protein, and hrTCS

Based on the above mentioned results, the four clinical parameters were arranged to a clinical prediction rule; the score’s items and the relative weightings chosen are listed in Table 2. 

The relative distribution of patients and the proportion of patients with a final diagnosis of cGCA in different score categories are shown in Figure 1. Given the very low prevalence of cGCA in patients with a score of 0 or 1 (3.8%), these score categories were classified as low clinical probability, whereas the score categories ≥ 2 (prevalence of cGCA 79.4%) were categorized as non-low clinical probability.

The proposed diagnostic algorithm suggests hrTCS for all patients with non-low clinical probability (score ≥ 2), irrespective of CRP values. Patients with low clinical probability (score < 2) are further stratified according to the CRP values. Within the diagnostic algorithm, a diagnosis of cGCA is rejected without sonographic imaging in patients with low clinical probability (score < 2) and CRP values below the cut-off of ≥2.5 mg/dL. In contrast, the algorithm assigns patients with low clinical probability and CRP values ≥ 2.5 mg/dL to undergo hrTCS (Figure 2).

### 3.4. Performance of the Diagnostic Algorithm in the Derivation Cohort

A cut-off of ≥2 divided the derivation cohort in 53 patients (60.9%) with low clinical probability and 34 patients (39.1%) with non-low clinical probability (Figure 3). According to the diagnostic algorithm, 25 patients with low clinical probability (score < 2) and CRP values ≥ 2.5 mg/dL were assigned to hrTCS (2 patients with a final diagnosis of cGCA, one of whom with a negative hrTCS study). Twenty-eight patients (32.2% of the overall cohort) with low clinical probability and CRP values < 2.5 mg/dL would not have been tested with sonographic imaging, and none of these patients had a final clinical diagnosis of cGCA. In the subgroup of 34 patients with non-low clinical probability, 24 patients were finally diagnosed with cGCA (21 patients with positive hrTCS) and 10 patients were classified as alternative diagnoses (two patients with positive hrTCS: both older males with anterior ischemic optic neuropathy (AION) of nonarteritic etiology and negative TAB; patients C1_a47 and C1_a50 in the Appendix A).

### 3.5. Performance of the Diagnostic Algorithm in the Validation Cohort

A cut-off of ≥2 divided the validation cohort in 60 patients (52.6%) with low clinical probability and 54 patients (47.4%) with non-low clinical probability (Figure 3). According to the diagnostic algorithm, only four patients with low clinical probability (score < 2) and CRP values > 2.5 mg/dL were assigned to hrTCS (one patient with a final diagnosis of cGCA who had a positive hrTCS study; three patients with a negative hrTCS study finally classified as alternative diagnoses). Fifty-six patients (49.1%) with low clinical probability and CRP values < 2.5 mg/dL would not have been tested with sonographic imaging, and none of these patients had a final clinical diagnosis of cGCA. In the subgroup of 54 patients with non-low clinical probability, 29 patients were finally diagnosed with cGCA (all with a positive hrTCS), and 25 patients were classified as alternative diagnoses (5 patients with positive hrTCS: 4 patients with nonarteritic AION, 1 patient with embolic central retinal artery occlusion; patients C2_a06, C2_a49, C2_a63, C2_a74, C2_a77 in the Appendix A).

### 3.6. Test Performance of hrTCS Dependent on Pre-Test Probability

The test performances of hrTCS in different point categories of the clinical prediction rule are summarized in Table 3 (both cohorts taken together). When applied in the derivation cohort on all patients with low clinical probability (*n* = 53), hrTCS exhibited a positive predictive value (PPV) and negative predictive value (NPV) of 50% and 98.4%, respectively. When applied only in patients with low clinical probability and CRP values of at least 2.5 mg/dL (*n* = 25), the PPV and NPV were 100% and 95.8%, respectively. In patients with non-low clinical probability (*n* = 34), hrTCS had a PPV and NPV of 91.3% and 72.7%, respectively. When applied in the validation cohort on all patients with low clinical probability (*n* = 60), hrTCS exhibited a PPV and NPV of 11.1% and 100%, respectively. Noteworthy, eight out of nine patients from the derivation cohort with low clinical probability but hrTCS values above the reference range were clinically classified as not having cGCA. Four patients with low clinical probability and CRP values of at least 2.5 mg/dL were correctly classified by hrTCS. In patients with non-low clinical probability (*n* = 54), hrTCS had a PPV and NPV of 85.3% and 100%, respectively. 

### 3.7. Extracranial GCA

Extracranial GCA was diagnosed in 16.1% and 5.2% of patients from cohort 1 and 2, respectively. While in cohort 1 half of the patients with extracranial GCA had no signs and symptoms of cGCA, all patients with extracranial GCA from cohort 2 had clear clinical and sonographic evidence of cGCA. Two patients with isolated extracranial GCA from cohort 1 would not have undergone hrTCS within the diagnostic algorithm.

## 4. Discussion 

Based on our results, we propose a simple clinical prediction rule for the diagnosis of cGCA. Our model is based on biographic information (age > 70 [12,15,17,18,21,25,26]), clinical symptoms which can be easily assessed by taking a careful medical history (headache [13,15,16,18,21,22], jaw claudication [12,13,14,15,16,17,18,21,22,24,25,26]), and a specific funduscopic finding obtained in patients presenting with visual impairment (AION [13]). New onset headache, not always restricted to the temple but sometimes manifesting as widespread headache, is the most common symptom of cGCA [33]. Older age and jaw claudication are established risk factors of permanent visual impairment in cGCA [34]. We used the item AION in our model, as AION is by far the most common cause of permanent visual impairment resulting from cGCA [35]. Our clinical prediction rule discriminated cGCA from alternative diagnoses with high diagnostic accuracy in the derivation cohort as well as in the independent validation cohort.

A stepwise diagnostic algorithm incorporating both the clinical prediction rule and a sensitive but nonspecific laboratory biomarker such as the CRP is helpful in identifying patients in whom a final clinical diagnosis of cGCA may even be excluded without imaging. According to the available evidence, we chose a CRP value of ≥2.5 mg/dL as threshold for performing hrTCS in patients stratified as having low risk in the clinical model [12,31,32]. One third of patients in the derivation cohort and almost half of the patients in the validation cohort were deemed to have low clinical probability according to the prediction rule and had CRP values below 2.5 mg/dL. None of these patients received a final diagnosis of cGCA. Such an approach may reduce resource utilization by reducing fast-track specialist referral for diagnostic imaging. Compared to a diagnostic strategy applying imaging in all suspected cases, this approach may also reduce the rate of false positive sonographic studies in patients with low clinical probability. Potential harms resulting from empiric high dose corticosteroid/biological treatment in false positive cases thus may be avoided in a considerable number of patients. 

Two patients finally diagnosed with isolated extracranial GCA from cohort 1 would not have been tested by imaging within the diagnostic algorithm. Both patients suffered from arm claudication and had typical findings of extracranial GCA in axillary artery CDS. In view of this finding, it must be noted that the clinical prediction rule is not intended to diagnose or exclude extracranial GCA. Given the highly variable clinical presentation of extracranial GCA [36], incorporation of symptoms of extracranial disease into a clinical prediction rule would result in a loss of simplicity, making the scoring system less useful for clinical practice. 

The diagnostic algorithm results in an enrichment of patients with a final diagnosis of cGCA in the category of patients classified as having a non-low clinical probability, thereby increasing the PPV of hrTCS in this subgroup. Few false positive findings remained in both cohorts analyzed, all in patients with a score of 2 or 3. Obviously, there remains a need for optimizing the sonographic protocol by means of establishing cut-off values of temporal artery wall thickness stratified according to age, sex, and cardiovascular risk profile [9,10]. Alternative ways of imaging assessment (e.g., considering a test as positive only in case of at least two affected temporal artery segments; semiquantitative scoring systems of temporal artery wall thickness) should be evaluated in this context [10,37].

Several clinical prediction rules have been built and tested for prediction of cGCA, mainly integrating clinical findings and laboratory markers to predict the probability of a positive TAB in ophthalmology settings [12,13,14,15,16,17,18]. Some authors provided calculators for prediction of the probability of cGCA [15,27]. Niederkohr et al. in 2005 proposed a decision analytic approach in order to give objective recommendations on whether or not to perform TAB uni- or bi-laterally [22]. In 2020, Monti and co-workers presented a combined clinical (cranial ischemic symptoms, polymyalgia rheumatica, and elevated acute phase reactants) and sonographic (intima media thickness above established reference values in the temporal arteries and axillary arteries, bilateral halo of the temporal arteries and axillary arteries) score based on data from the Role of Ultrasound compared to Biopsy in the Diagnosis and Treatment of GCA (TABUL) study. The score allowed identification of patients with a positive TAB with good diagnostic accuracy (AUC 0.77) [30]. El-Dairi et al. and Laskou et al. proposed diagnostic algorithms based on clinical probability scores derived from single center cohorts [25,28]. Of note, Laskou et al. were the first to use a clinical reference diagnosis, taking into account the clinical course over time [28]. Subsequently, this Southend pretest probability score, stratifying fast-track referral patients into low-risk, intermediate-risk, and high-risk probability categories based on 17 items, was shown to enhance the test performance of temporal/axillary artery sonography [29]. Most recently, a Canadian study contextualized the diagnostic yield of temporal artery magnetic resonance angiography in the diagnosis of GCA dependent on pre-test-probability [38]. However, to the best of our knowledge, a formal stepwise algorithm based on a clinical prediction rule for the diagnostic management of cGCA has not been established and validated to date. A positive effect of clinical prediction rules on process outcomes has been documented for various medical conditions, (e.g., reduction of additional diagnostic testing, early discharge, and symptom improvement) [39].

We acknowledge some limitations of our study. Both cohorts were retrospectively analyzed, and the final diagnosis of cGCA vs. alternative diagnoses was based on clinical judgement and not on histology of the temporal arteries in most of our cases. TAB was performed only in a minority of patients, reflecting the paradigm change towards an imaging-based diagnostic approach that took place in the last decade. The validation cohort consisted of preselected patients suffering from acute-onset, permanent visual impairment. Since 2014, we have been using hrTCS in addition to CDS of the temporal arteries for the diagnostic workup of suspected cGCA. Whether our results can be transferred to other study populations and to other sonographic approaches (CDS only, intima-media-thickness measurements of the cranial arteries) remains to be investigated.

## 5. Conclusions

In summary, we propose a simple clinical prediction rule, which, integrated into a stepwise algorithm, exactly discriminates patients with cGCA from patients with alternative diagnoses. This strategy implies a significant increase in diagnostic confidence both in primary and in specialist care when evaluating a patient with low clinical probability and only slightly elevated CRP values. Provided that the diagnostic yield of our diagnostic algorithm can be confirmed in prospective validation cohorts, a prospective management study to verify the safety and efficacy of the algorithm seems to be justified.

## Figures and Tables

**Figure 1 jcm-10-01163-f001:**
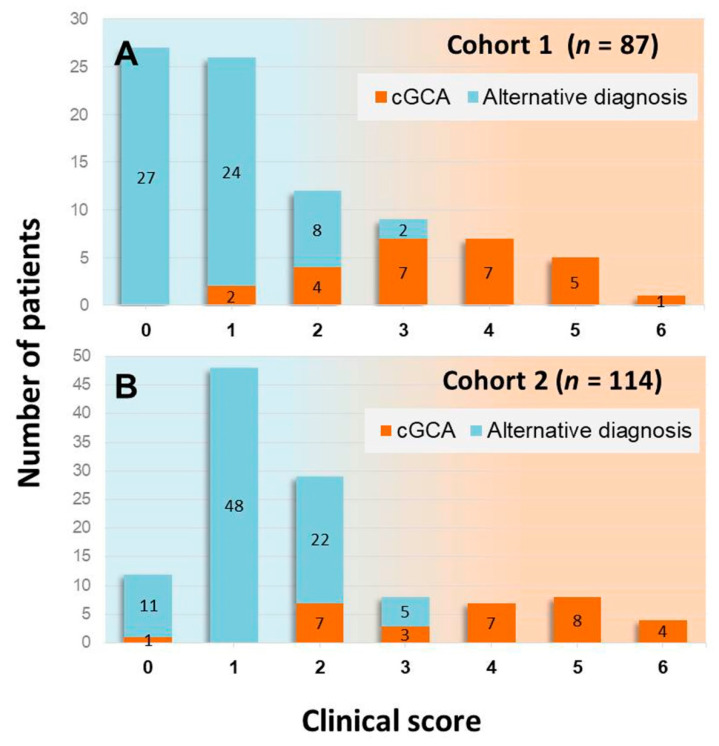
Prevalence of patients with and without a final diagnosis of cGCA in different point categories of the clinical score in the derivation cohort (**A**) and the validation cohort (**B**).

**Figure 2 jcm-10-01163-f002:**
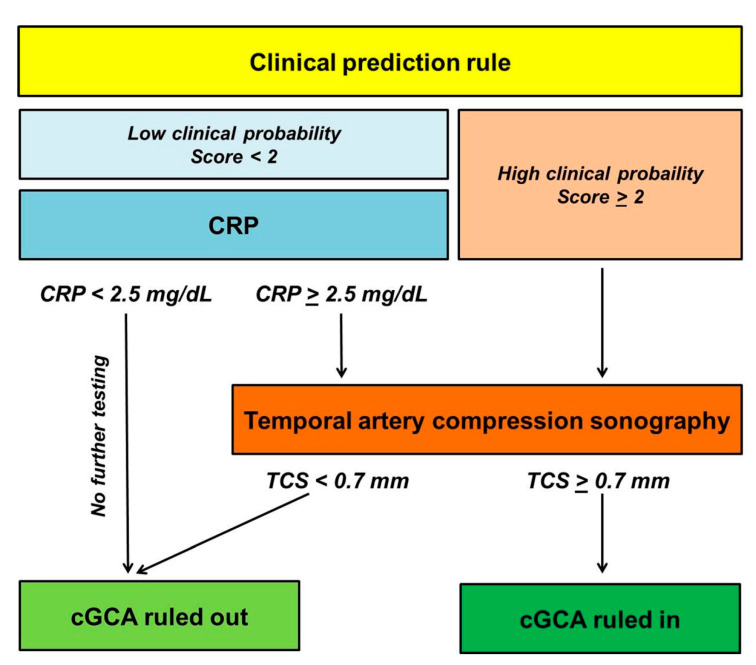
Setup of the proposed diagnostic algorithm integrating the clinical score, a CRP cut-off of ≥2.5 mg/dL in patients with low clinical probability, and high-resolution compression sonography (hrTCS) in patients with low clinical probability exhibiting CRP values above the cut-off of ≥2.5 mg/dL, as well as in all patients with non-low clinical probability regardless of CRP values. Patients with low clinical probability and a CRP value < 2.5 mg/dL are not assigned to undergo sonographic imaging. CRP, C-reactive protein, cGCA, cranial giant cell arteritis.

**Figure 3 jcm-10-01163-f003:**
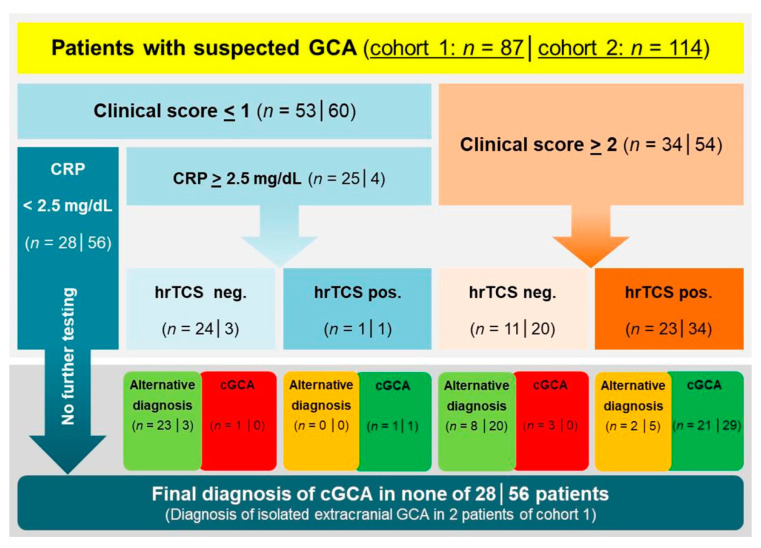
Discriminatory value of the diagnostic algorithm in both cohorts. Figures given in each box refer to the number of patients in the derivation cohort (left side) and the validation cohort (right side), respectively. CRP, C-reactive protein, GCA, giant cell arteritis, cGCA, cranial giant cell arteritis, hrTCS, high-resolution compression sonography.

**Table 1 jcm-10-01163-t001:** Comparison of clinical characteristics of patients with and without a final diagnosis of cranial giant cell arteritis (cGCA) in both cohorts.

	Cohort 1, Final Diagnosis of cGCA *n* = 26	Cohort 1, Final Diagnosis Not cGCA *n* = 61	Cohort 2, Final Diagnosis of cGCA *n* = 30	Cohort 2, Final Diagnosis Not cGCA *n* = 84
Age, years (mean ± SD)	73.2 (9.2)	66.1 (11.2)	77.5 (6.7)	73.6 (10.2)
Female sex (*n*, %)	15 (57.7)	33 (54.1)	19 (63.3)	40 (47.6)
New onset headache (*n*, %)	21 (80.8)	11 (18)	19 (63.3)	13 (15.5)
Jaw claudication (*n*, %)	16 (61.5)	2 (3.3)	18 (60)	0
Amaurosis fugax (*n*, %)	4 (15.4)	7 (11.5)	2 (6.7)	4 (4.8)
Permanent sight loss (*n*, %)	19 (73.1)	18 (29.5)	30 (100)	84 (100)
AION (*n*, %)	16 (61.5)	7 (11.4)	28 (93.3)	26 (31)
Bilateral AION (*n*, %)	3 (11.5)	1 (1.6)	6 (20)	3 (3.6)
PMR (*n*, %)	10 (38.5)	21 (34.4)	6 (20)	1 (1.2)
Constitutional symptoms (*n*, %)	12 (46.2)	19 (31.1)	13 (43.3)	4 (4.8)
CRP (mg/dL, mean ± SD)	5.2 (5.3)	4.2 (5.6)	5.1 (5.7)	0.8 (0.9)
TAB performed (*n*, %)	13 (50)	6 (9.8)	8 (26.7)	9 (10.7)
TAB positive (*n*, %)	10 (38.5)	0	5 (16.7)	0

AION, anterior ischemic optic neuropathy; CRP, C-reactive protein; PMR, polymyalgia rheumatica; TAB, temporal artery biopsy.

**Table 2 jcm-10-01163-t002:** Clinical prediction model, derived from cohort 1.

Variable	Description	Score
Age (years)	<70 years	0
	>70 years	1
New onset persistent headache	No	0
	Yes	1
Jaw claudication	No	0
	Yes	1
Permanent vision impairment due to AION	No	0
	Unilateral	1
	Bilateral	2
Score (range 0–6)	Low clinical probability	≤1 point
	High clinical probability	≥2 points

AION, anterior ischemic optic neuropathy.

**Table 3 jcm-10-01163-t003:** Positive and negative predictive value of hrTCS in different score categories (both cohorts together, *n* = 201).

Score	Proportion of Patients (*n*, %)	Prevalence of cGCA (*n*, %)	PPV	NPV
0	39 (18.8)	1 (2.6)	50	100
1	74 (35.7)	2 (2.7)	11.1	98.4
2	39 (18.8)	11 (28.2)	69.2	92.3
3	17 (8.2)	10 (58.8)	75	80
4	14 (6.8)	14 (100)	100	/ *
5	13 (6.3)	13 (100)	100	/ *
6	5 (2.4)	5 (100)	100	/ *

* In both cohorts, all patients with a score of 4–6 had a positive hrTCS study and were finally diagnosed with cGCA. PPV, positive predictive value, NPV, negative predictive value.

## Data Availability

The data that support the findings of this study are available from the corresponding author upon reasonable request.

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
