# Peer review of "A Diagnostic Algorithm Based on a Simple Clinical Prediction Rule for the Diagnosis of Cranial Giant Cell Arteritis"

_jcm, 2021, doi:10.3390/jcm10061163_

Round 1
Reviewer 1 Report
It is a valuable study assessing a new algorithm for GCA diagnosis prudently aiming to “keep the clinical model as simple as possible”. I find no serious caveats.
Some minor language mistakes need awareness (e.g. stratification instead of „Tratification” hrtCS or hrTCS).
Was “alternative diagnoses” defined as non-GCA, some unknown cause for the patients’ presentations, or was the alternative disease exactly stated?
Author Response
We thank Reviewer 1 for his comments. Please find the point-by-point-Reply below.
(1) "Some minor language mistakes need awareness (e.g. stratification instead of „Tratification” hrtCS or hrTCS)."
Minor language issues were corrected.
(2) "Was “alternative diagnoses” defined as non-GCA, some unknown cause for the patients’ presentations, or was the alternative disease exactly stated?"
Alternative disease entities were recorded. We added a supplementary file including the exact alternative diagnoses in both cohorts.
Reviewer 2 Report
Dear Authors,
here are my comments and suggestions:
- You proposed a diagnostic algorithm for the diagnosis of cranial GCA (cGCA). Is CDS of the axillary arteries recommended as the first imaging test (please, see line3-5 in Introduction section) ? No. Therefore, this Your sentence should be modified.
- GCA without cranial manifestations has been proposed as a distinct subset of disease. Please, clarify the “boundaries” of Your reasearch disegn in Introduction, and discuss this point (in the Discussion section). Besides, You reported patients with extracranial GCA in 3.7 paragraph. Why ? You stated that You was only concerned with the cranial GCA. My personal opinion is that this part should be removed because it can be misleading.
- You didn’t discuss the relationship between GCA and polymyalgia rheumatica (PMR). Had all Your enrolled patients GCA only, without PMR ? Please, clarify this point.
- In Introduction, You wrote “In clinical practice........several clinical prediction rules have been developed and in part validated in order to perform explicit (structured) pre-test probability assessment....”. I think that You should quote these clinical prediction rules (or others....) and add their references. Otherwise, this Your statement is too generic.
- 3.3. Stepwise diagnostic algorithm integrating the clinical model, C-reactive protein and hrTCS. You wrote “In patients with CRP-values below the cut-off of > 2.5 mg/dl a diagnosis of cGCA is rejected”. Please, clarify this point. Indeed, in line with the diagnostic criteria You considered, a diagnosis of GCA is possible when 3/5 proposed criteria are present. Furthermore, diagnosis of biopsy-proven GCA is possible without CRP increase.
- 4. Performance of the diagnostic algorithm in the derivation cohort : In the subgroup of 34 patients with non-low clinical probability, 3 patients with positive hrTCS has an alternative diagnosis. 3.5. Performance of the diagnostic algorithm in the validation cohort: in the subgroup of 54 patients with non-low clinical probability .....5 with positive hrTCS had alternative diagnoses. Please, clarify these points : what were these alternative diagnoses ?
- In Discussion section, the part from “Only limited data suggest.....” to “and a sonographic imaging test as the diagnostic standard” should be removed because it was unnecessary.
Further references should be considered. In particular, I suggest: de Boysson et al, 2016. doi: 10.1097/MD.0000000000003818 and Mollan et al, 2020. doi: 10.1186/s10194-020-01093-7. Besides, Your literature review (see 2.2 Derivation of the clinical model) seems not complete. For instance, I don’t found Manzo, 2016. doi: 10.5114/reum.2016.63663, and Hocevar et al, 2020. doi: 10.1093/rheumatology/keaa058. Please, add these references. Take into account also what I wrote in point no. 4 .
Author Response
We thank Reviewer 1 for his comments. Please find our point-by-point-peply below:
(1) "You proposed a diagnostic algorithm for the diagnosis of cranial GCA (cGCA). Is CDS of the axillary arteries recommended as the first imaging test (please, see line3-5 in Introduction section) ? No. Therefore, this Your sentence should be modified."
Done. Please see revised Section 1.
(2) "GCA without cranial manifestations has been proposed as a distinct subset of disease. Please, clarify the “boundaries” of Your reasearch disegn in Introduction, and discuss this point (in the Discussion section). Besides, You reported patients with extracranial GCA in 3.7 paragraph. Why ? You stated that You was only concerned with the cranial GCA. My personal opinion is that this part should be removed because it can be misleading."
Unfortunately, disease patterns of cranial and extracranial GCA overlap in a substantial number of patients. While isolated cranial GCA carries a high risk of acute ischemic complications (blindness), the extracranial disease pattern is associated with a lower risk of blindness. Given this overlap, we would prefer to leave the statement on extracranial GCA in the results section if this is somehow possible. The discussion part already contains a statement on the limitations of the prediction rule in detection of extracranial GCA. We added a sentence on the boundaries of our study aims to the Introduction (please see revised Section 1).
(3) "You didn’t discuss the relationship between GCA and polymyalgia rheumatica (PMR). Had all Your enrolled patients GCA only, without PMR ? Please, clarify this Point."
The rate of patients presenting with symptoms of PMR in both cohorts is given in table 1. As PMR was not included in the final model, we did not discuss the well known association between GCA and PMR further.
(4) In Introduction, You wrote “In clinical practice........several clinical prediction rules have been developed and in part validated in order to perform explicit (structured) pre-test probability assessment....”. I think that You should quote these clinical prediction rules (or others....) and add their references. Otherwise, this Your statement is too generic.
We fully agree and provide the respective references in the revised manuscript.
(5) 3.3. Stepwise diagnostic algorithm integrating the clinical model, C-reactive protein and hrTCS. You wrote “In patients with CRP-values below the cut-off of > 2.5 mg/dl a diagnosis of cGCA is rejected”. Please, clarify this point. Indeed, in line with the diagnostic criteria You considered, a diagnosis of GCA is possible when 3/5 proposed criteria are present. Furthermore, diagnosis of biopsy-proven GCA is possible without CRP increase.
This is an important comment, revealing the need for clarification in our manuscript. The quoted sentence refers to the proposed diagnostic algorithm, whereas the mentioned diagnostic criteria were used for establishing the clinical reference diagnosis against which the novel diagnostic algorithm was validated. We agree that wording in this context was not optimal and we clarified this in the revised manuscript (please see revised Section 3.3).
(6) "Performance of the diagnostic algorithm in the derivation cohort : In the subgroup of 34 patients with non-low clinical probability, 3 patients with positive hrTCS has an alternative diagnosis. 3.5. Performance of the diagnostic algorithm in the validation cohort: in the subgroup of 54 patients with non-low clinical probability .....5 with positive hrTCS had alternative diagnoses. Please, clarify these points : what were these alternative diagnoses ?"
We added the information on alternative diagnoses in patients with non-low clinical probability and a positice hrTCS-study from both cohorts to the main text. Furthermore, together with the revised manuscript we provide a supplementary table including complete information on the specific alternative disease entities in patients without a final clinical diagnosis of GCA (please see revised Section 3.4, 3.5 and the Supplementary table).
(7) "In Discussion section, the part from “Only limited data suggest.....” to “and a sonographic imaging test as the diagnostic standard” should be removed because it was unnecessary."
Done.
(8) Further references should be considered. In particular, I suggest: de Boysson et al, 2016. doi: 10.1097/MD.0000000000003818 and Mollan et al, 2020. doi: 10.1186/s10194-020-01093-7. Besides, Your literature review (see 2.2 Derivation of the clinical model) seems not complete. For instance, I don’t found Manzo, 2016. doi: 10.5114/reum.2016.63663, and Hocevar et al, 2020. doi: 10.1093/rheumatology/keaa058. Please, add these references. Take into account also what I wrote in point no. 4 .
Done (see also response to point 4).
Round 2
Reviewer 2 Report
Dear Authors,
I appreciate the changes You made. However, not all my comments and suggestions were satisfactorily met.
Specifically, the paragraph 3.7 is still misleading, and not all the proposed references have been taken into account. For instance, I do not found in Your newer version this reference: Manzo C. Widespread headache as first manifestation of giant cell arteritis.....". Reumatologia 2016; 54:236-238.
Please revise this paragraph and add this reference.
Author Response
Dear Editor, Dear Reviewer,
please find enclosed the second revision of our manuscipt.We completely re-wrote and significantly shortened paragraph 3.7. and added the suggested reference. We hope that our paper now meets the scientific standards of the Journal of Clinical Medicine. We are looking forward to the Editorial decision.
Yours sincerely
Michael Czihal M.D. (on behalf of all authors)